# Identification of Spinal Inhibitory Interneurons Required for Attenuating Effect of Duloxetine on Neuropathic Allodynia-like Signs in Rats

**DOI:** 10.3390/cells11244051

**Published:** 2022-12-14

**Authors:** Tadayuki Ishibashi, Daichi Sueto, Yu Yoshikawa, Keisuke Koga, Ken Yamaura, Makoto Tsuda

**Affiliations:** 1Department of Molecular and System Pharmacology, Graduate School of Pharmaceutical Sciences, Kyushu University, Fukuoka 812-8582, Japan; 2Department of Anesthesiology and Critical Care Medicine, Graduate School of Medical Sciences, Kyushu University, Fukuoka 812-8582, Japan; 3Department of Neurophysiology, Hyogo College of Medicine, Nishinomiya 663-8501, Japan; 4Kyushu University Institute for Advanced Study, Fukuoka 819-0395, Japan

**Keywords:** neuropathic pain, allodynia, noradrenaline, duloxetine, spinal dorsal horn interneurons

## Abstract

Neuropathic pain is a chronic pain condition that occurs after nerve damage; allodynia, which refers to pain caused by generally innocuous stimuli, is a hallmark symptom. Although allodynia is often resistant to analgesics, the antidepressant duloxetine has been used as an effective therapeutic option. Duloxetine increases spinal noradrenaline (NA) levels by inhibiting its transporter at NAergic terminals in the spinal dorsal horn (SDH), which has been proposed to contribute to its pain-relieving effect. However, the mechanism through which duloxetine suppresses neuropathic allodynia remains unclear. Here, we identified an SDH inhibitory interneuron subset (captured by adeno-associated viral (AAV) vectors incorporating a rat *neuropeptide Y* promoter; AAV-NpyP^+^ neurons) that is mostly depolarized by NA. Furthermore, this excitatory effect was suppressed by pharmacological blockade or genetic knockdown of α_1B_-adrenoceptors (ARs) in AAV-NpyP^+^ SDH neurons. We found that duloxetine suppressed Aβ fiber-mediated allodynia-like behavioral responses after nerve injury and that this effect was not observed in AAV-NpyP^+^ SDH neuron-selective α_1B_-AR-knockdown. These results indicate that α_1B_-AR and AAV-NpyP^+^ neurons are critical targets for spinal NA and are necessary for the therapeutic effect of duloxetine on neuropathic pain, which can support the development of novel analgesics.

## 1. Introduction

Neuropathic pain develops due to a lesion or disease of the somatosensory system [1]. Mechanical allodynia, which refers to pain caused by otherwise innocuous mechanical stimuli, is a symptom of neuropathic pain. Mechanical information from the skin is transmitted to the spinal dorsal horn (SDH) via low-threshold mechanoreceptors such as Aβ fibers [2]. Normally, Aβ fibers do not activate brain-projecting pain transmission neurons located in lamina I of the SDH, but after peripheral nerve injury (PNI), activation occurs and pain is induced (neuropathic allodynia) [3,4]. An increasing body of evidence indicates that dysfunction of inhibitory interneurons in the SDH after PNI critically contributes to Aβ fiber-mediated excitation of nociceptive lamina I neurons and neuropathic allodynia [5,6,7,8,9]. Thus, the enhancement of inhibitory interneuron activity would be an effective way to mitigate neuropathic allodynia.

Somatosensory information processing in the SDH is regulated by serotonin and noradrenaline (NA) released from spinal terminals of descending neurons in the brain [10,11,12]. The descending NAergic pathway has been proposed as a target of duloxetine [13,14,15], a serotonin and NA reuptake inhibitor (SNRI) that has shown clinical efficacy for treating neuropathic pain [1,16]. Duloxetine increases spinal NA levels by inhibiting its transporters at NAergic terminals [17,18,19]. Within the SDH, NA inhibits glutamate release from the presynaptic terminal of C fibers and excitation of SDH interneurons via α_2_-adrenoceptors (ARs) [20]. However, the role of α_2_-ARs in the duloxetine-mediated mitigation of neuropathic pain remains controversial. The effect of duloxetine is reportedly reduced by yohimbine, an α_2_-AR antagonist; however, the selectivity ratio between α_2_-AR and α_1_-AR is much lower than that of the α_2_-AR-selective antagonist idazoxan [21]. It has also been shown that idazoxan reduces the suppressive effect of duloxetine repeatedly administered on neuropathic behavioral hypersensitivity [19] but not that of acutely administered duloxetine [22]. In addition to α_2_-AR, NA also activates α_1_-AR and has an excitatory effect on inhibitory interneurons in the SDH [23,24,25,26]. However, the role of duloxetine in attenuating neuropathic pain-like behavior is poorly understood. Furthermore, despite recent advances in our understanding of the heterogeneity of inhibitory interneurons in the SDH [7,27,28], the neuronal subset that targets spinal NA necessary for the effect of duloxetine remains unclear.

In the present study, we investigated the mechanism by which duloxetine modulates neuropathic pain by focusing on an SDH inhibitory interneuron subset (captured by an adeno-associated viral [AAV] vector including a rat *neuropeptide Y* promoter; AAV-NpyP^+^ neurons), which has previously been identified as a key player for Aβ fiber-evoked allodynia-like behavior after PNI [9]. Using multiple approaches (electrophysiology, subset-selective RNA interference, and optogenetics), we identified AAV-NpyP^+^ SDH neurons as critical targets of spinal NA that contribute to the analgesic effect of duloxetine on neuropathic allodynia.

## 2. Materials and Methods

### 2.1. Animals

W-Tg (Thy1-COP4/YFP*)4Jfhy (W-TChR2V4: NBRPRat No. 0685) rats [29,30] were supplied by the National BioResource Project—Rat, Kyoto University (Kyoto, Japan). All male wild-type (WT) and W-TChR2V4 rats were aged 4–5 weeks at the start of each experiment. Rats were housed in individual cage (21–23 °C, light on 8:00 to 20:00) and were fed food and water ad libitum. Rats were randomly allocated to each experimental group.

### 2.2. Recombinant Adeno-Associated Virus (rAAV) Vector Production

To produce rAAV vector for *Npy* promoter-dependent gene transduction, a vector containing the *Npy* promoter (NCBI GenBank: HM443071.1; 1659 bp; –1633 to +26 (+1 = transcription start site)) was generated from pZac2.1 by substituting the CMV promoter with the *Npy* promoter [9]. tdTomato (tdT) was cloned into the modified pZac2.1 to generate pZac2.1-NpyP-tdT-WPRE. This rAAV vector was produced from human embryonic kidney 293T (HEK293T) cells with triple transfection (pZac, cis plasmid; pAAV2/9, trans plasmid; pAd DeltaF6, adenoviral helper plasmid (all plasmids were purchased from the University of Pennsylvania Gene Therapy Program Vector Core)) and purified by two CsCl density gradient purification steps. The vector was dialyzed against phosphate-buffered saline (PBS, Wako, Osaka, Japan) containing 0.001% (*v*/*v*) Pluronic-F68 using Amicon Ultra 100K filter units (Millipore, Darmstadt, Germany). The genome titer of rAAV was measured by Pico Green fluorometric reagent (Molecular Probes, Eugene, OR, USA) after denaturation of the AAV particle. Vectors were stored in aliquots at −80 °C until use.

### 2.3. In Vivo AAV-NpyP^+^ Neurons-Specific Knockdown with AAV Vector Encoding shRNA

The gene encoding shRNAmir (from pGIPZ-mir30-shGapdh, Thermo Fisher Scientific, Waltham, MA, USA) and mCherry were subcloned into pENTR plasmid. Synthetic oligonucleotides including targeting sequence for *Adra1b* (GCGGGAGTCATGAAGGAAATG) and non-targeting sequence (GATTCGACATCATGTATCTT) were replaced with the targeting site in the original pENTR-mCherry-mir30. The resulting mCherry-mir30-shRNA cassette was transferred into the AAV shuttle vector (pZac2.1-NpyP-mCherry-shmirAdra1b-WPRE and pZac2.1-NpyP-mCherry-shmirControl-WPRE).

### 2.4. Intra-SDH Injection of rAAV Vector

According to our previous methods [9,31], rats were intraperitoneally (i.p.) injected with pentobarbital (65 mg/kg) under isoflurane (2%) anesthesia. In brief, we inserted the microcapillary backfilled with rAAV solution into the SDH between Th13 and L1 vertebrae (500 μm lateral from the midline and 250 μm in depth from the surface of the dorsal root entry zone) and microinjected 800 nl rAAV solution (3 × 10^12^ GC/mL) using FemtoJet Express (Eppendorf, Hamburg, Germany). After microinjection, we slowly removed the inserted microcapillary from the SDH and the skin was sutured with 3–0 silk.

### 2.5. Immunohistochemistry

To anesthetize rats, pentobarbital (100 mg/kg, i.p.) was administered. Fully anesthetized rats were perfused transcardially with PBS, followed by ice-cold 4% paraformaldehyde/PBS. The L4 segments of the spinal cord were removed, postfixed in the same fixative for 3 h at 4 °C, placed in 30% sucrose solution for 24–48 h at 4 °C, embedded by Tissue-Tek O.C.T. Compound (Sakura Finetek Japan, Tokyo, Japan) and stored at −80 °C before use. The L4 spinal cord transverse sections (30 μm) were made and were immunostained by the free-floating method as described previously [9,32]. The sections were incubated in blocking solution for 2 h and followed by the primary antibodies: polyclonal goat anti-paired box 2 (PAX2; 1:500; AF3364, R&D Systems, Minneapolis, MN, USA) and isolectin B4 (IB4) biotin-conjugate (1:1000; I21414, Thermo Fisher Scientific, Waltham, MA, USA) for 48 h at 4 °C. The sections were washed and incubated with secondary antibodies conjugated with streptavidin-conjugated Alexa Fluor 405 and Alexa Fluor 488 (1:1000; S32351 and A11055, Molecular Probes, Eugene, OR, USA) for 3 h. We then analyzed the sections using a confocal microscope (LSM700, Zeiss, Oberkochen, Germany).

### 2.6. Whole-Cell Patch-Clamp Recordings

As previously described [4,9], under anesthesia with urethane (1.2–1.5 g/kg, i.p.), the lumbosacral spinal cord was removed from WT rats and placed into a cold high-sucrose artificial cerebrospinal fluid (aCSF) (250 mM sucrose, 2.5 mM KCl, 2 mM CaCl_2_, 2 mM MgCl_2_, 1.2 mM NaH_2_PO_4_, 25 mM NaHCO_3_, and 11 mM glucose). Using a vibrating microtome (Leica VT1200, Leica), a transverse spinal cord slice (350–400 µm thick) was made, and was maintained in oxygenated aCSF solution (125 mM NaCl, 2.5 mM KCl, 2 mM CaCl_2_, 1 mM MgCl_2_, 1.25 mM NaH_2_PO_4_, 26 mM NaHCO_3_, 20 mM glucose, 10 μM bicuculline, and 50 μM D-(−)-2-amino-5-phosphonovaleric acid) at 22–25 °C (room temperature) for at least 30 min. We put the spinal cord slice into a recording chamber where oxygenated aCSF solution (26–28 °C) was continuously superfused at a flow rate of 4–7 mL/min. SDH neurons were visualized with an upright microscope equipped with infrared differential interference contrast Nomarski (FN1, Nikon, Tokyo, Japan). The patch pipettes were filled with an internal solution (125 mM K-gluconate, 10 mM KCl, 0.5 mM EGTA, 10 mM HEPES, 4 mM ATP-Mg, 0.3 mM NaGTP, 10 mM phosphocreatine, pH 7.28 adjusted with KOH). The pipette tip resistance was 4–7 MΩ. Using a computer-controlled amplifier (Axopatch 700B, Molecular Devices, San Jose, CA, USA), membrane potentials were recorded. The data were digitized with an analog-to-digital converter (Digidata 1550, Molecular Devices, San Jose, CA, USA), stored on a personal computer using a data acquisition program (pCLAMP 10.4 acquisition software, Molecular Devices, San Jose, CA, USA), and analyzed using a software package (Clampfit version 10.7, Molecular Devices, San Jose, CA, USA). Resting membrane potentials (RMPs) were recorded in current-clamp mode. The drugs used were L-norepinephrine hydrochloride (NA; 20 μmol/L, Sigma-Aldrich, St. Louis, MO, USA), silodosin (30 nmol/L, Wako, Osaka, Japan), L-765,314 (1 μmol/L, Santa Cruz Biotechnology, Dallas, TX, USA), A-315456 (1 μmol/L, Sigma-Aldrich, St.Louis, MO, USA), and atipamezole hydrochloride (1 μmol/L, Wako, Osaka, Japan). All drugs were dissolved in aCSF solution. NA was superfused for 2 min. Each antagonist was pre-superfused for 3 min before NA and later was co-perfused with NA for 2 min. We quantified averaged RMP for 1 min of pre-drug and post-drug application, and a change in RMP (ΔRMP) of 5 mV or more was judged to be depolarization or hyperpolarization.

### 2.7. Neuropathic Pain Model

We used the spinal nerve injury model with some modifications [32,33]. In brief, under isoflurane (2%) anesthesia, the L5 spinal nerve was tightly ligated with 5–0 silk and cut just distal to the ligature. The wound and the surrounding skin were sutured with 3–0 silk.

### 2.8. Light Illumination of Hind paw

According to our previous methods [4,9], we placed rats on a transparent acrylic plate and habituated them for 30–60 min. The plantar surface of the hind paw (touching the acrylic plate floor) was illuminated with a blue laser diode (COME2-LB473/532/100, Lucir, Osaka, Japan: wavelength, 470 nm; frequency, 5 Hz; interval, 10 s; 10 times per each ipsilateral and contralateral hind paw) over the acrylic plate. Using a thermopile (COME2-LPM-NOVA, Lucir, Osaka, Japan), we measured the light power intensity (1 mV), which was a laser power meter with 1 mW at the skin. Withdrawal responses of the hind paw to light illumination (0: no reaction, 1: mild movement without any lifting and flinching behaviors, 2: hind paw lifting and flinching) were calculated from total scores of 10 times per hind paw. To investigate the light-induced responses of animals, we were careful to establish experimental conditions prior to light stimuli. First, the animals were awake. Second, both hind paws were clearly attached to the floor of the acrylic plate. Third, the animals were at rest without moving or walking.

### 2.9. Von Frey Test

As previously described [4,9], calibrated von Frey filaments (0.4–15 g, Stoelting, Wood Dale, IL, USA) were applied to the plantar surface of the rat hind paw, and the 50% paw withdrawal threshold was determined.

### 2.10. Intraperitoneally Administration of Duloxetine

Two weeks after PNI, W-TChR2V4 rats were intraperitoneally administrated duloxetine (30 mg/kg) dissolved by saline.

### 2.11. Statistical Analysis

Statistical analyses of the results were conducted with Fisher’s exact test followed by Bonferroni adjustment (Figure 1C), Fisher’s exact test (Figure 2B), two-way repeated measures ANOVA with post hoc Bonferroni’s multiple comparison test (Figure 3A), and two-way repeated measures ANOVA with post hoc Tukey’s multiple comparison test (Figure 3B) using GraphPad Prism 7.01 software (GraphPad software Inc., San Diego, CA, USA). *p* values are indicated as * *p* < 0.05, ** *p* < 0.01, *** *p* < 0.001, and **** *p* < 0.0001, or ^#^
*p* < 0.05, ^##^
*p* < 0.01, ^###^
*p* < 0.001, and ^####^
*p* < 0.0001. Data are shown as the mean ± SEM (Figure 2A, Figure 3A,B).

## 3. Results

### 3.1. NA Excites the Majority of AAV-NpyP^+^ Neurons Via α_1B_-AR

To examine the activity of AAV-NpyP^+^ SDH neurons, we visualized AAV-NpyP^+^ neurons by intra-SDH microinjection of the AAV-NpyP vector including the gene encoding tdTomato (tdT) in wild-type (WT) rats (Figure 1A). AAV-NpyP^+^ neurons (tdT^+^ cells) were located in lamina IIo and immunolabeled with paired box 2 (PAX2) as a marker of inhibitory neurons (Figure 1A). Using spinal cord slices from tdT-expressing WT rats, whole-cell recordings from AAV-NpyP^+^ neurons were performed under the current clamp mode. We measured averaged resting membrane potential (RMP) for 1 min of pre-drug and post-drug application, and a change in RMP (ΔRMP) of 5 mV or more was judged to be depolarization or hyperpolarization. NA application produced depolarization in 64.3% of the AAV-NpyP^+^ neurons (*n* = 18/28), and more than half of the depolarizing neurons (*n* = 11/18) evoked action potential firing (Figure 1B,C). The remaining neurons (*n* = 9/28) exhibited hyperpolarization (Figure 1B,C). The average changes in the RMPs of NA-depolarizing and hyperpolarizing neurons were 12.43 and −7.67 mV, respectively (Figure 1D). These data indicate that almost all AAV-NpyP^+^ neurons responded to NA, with depolarization being the predominant response.

### 3.2. Knockdown of α_1B_-AR in AAV-NpyP^+^ Neurons Suppresses NA-Evoked Depolarization

To determine the AR subtype responsible for the effects of NA, spinal slices were treated with subtype-specific antagonists. NA-induced depolarization in AAV-NpyP^+^ neurons was prevented by the α_1B_-AR antagonist L-765,314 (Figure 1C) but not by silodosin (α_1A_-AR) or A-315456 (α_1D_-AR). When L-765,314 was treated, the proportion of AAV-NpyP^+^ neurons exhibiting hyperpolarization increased, and co-treatment with atipamezole (α_2_-AR antagonist) abolished the NA-induced responses (Figure 1C,D). To determine the role of α_1B_-ARs expressed in AAV-NpyP^+^ neurons, short hairpin RNA (shRNA) targeting α_1B_-ARs (shmirAdra1b) was used to knockdown gene expression in AAV-NpyP^+^ neurons by intra-SDH microinjection of AAV-NpyP-mCherry-shmirAdra1b. We confirmed mCherry expression in the PAX2^+^ lamina IIo neurons, which is consistent with our previous findings [9] (Figure 2A). Electrophysiological recordings revealed that the NA failed to depolarize almost all mCherry^+^ neurons (Figure 2B,C). These results indicated that NA-induced depolarization and hyperpolarization are mediated by α_1B_-ARs and α_2_-ARs, respectively. Furthermore, AAV-NpyP^+^ neurons depolarized by NA would also express α_2_-ARs, but the net effect of NA in these neurons is excitatory.

### 3.3. Duloxetine Alleviates Neuropathic Allodynia-like Behavior Via α_1B_-AR in AAV-NpyP^+^ Neurons

As duloxetine has been shown to increase spinal NA [17,18,19], we predicted that α_1B_-ARs in AAV-NpyP^+^ neurons could be involved in its effect on neuropathic pain. To determine this, we examined the effect of duloxetine on neuropathic allodynia using transgenic rats (W-TChR2V4) expressing channelrhodopsin-2 (ChR2) at the nerve endings of touch-sensing Aβ fibers [4,30]. We found that the pain-like withdrawal behavior elicited by photostimulation of Aβ fibers by applying light to the plantar skin 2 weeks after PNI was suppressed by intraperitoneal administration of duloxetine (30 mg/kg) (Figure 3A). A similar effect was observed in PNI-induced behavioral hypersensitivity to mechanical stimulation by von Frey filaments (Figure 3A). We further found that the suppressive effects of duloxetine were abolished in AAV-NpyP^+^ neuron-specific α_1B_-AR-knockdown (Figure 3B). In addition, we confirmed that the excitatory effect of NA on AAV-NpyP^+^ neurons was retained after PNI, and in spinal slices from WT rats 2 weeks after PNI; NA caused depolarization of 65.0% of AAV-NpyP^+^ neurons (*n* = 13/20), and the average change in RMPs was 10.73 mV. These results indicate that α_1B_-ARs expressed in AAV-NpyP^+^ neurons are necessary for the analgesic effect of duloxetine on neuropathic allodynia-like behavior.

## 4. Discussion

In this study, we identified AAV-NpyP^+^ SDH neuron subset that predominantly exhibited depolarization to spinal NA via α_1B_-ARs. We further found that the suppressive effect of duloxetine on Aβ fiber-mediated allodynia-like behavioral responses of a model of neuropathic pain was not observed in AAV-NpyP^+^ SDH neuron-selective α_1B_-AR-knockdown. From these findings, this study demonstrates that α_1B_-AR and AAV-NpyP^+^ neurons are critical targets for spinal NA and are necessary for the therapeutic effect of duloxetine on neuropathic pain. Thus, α_1B_-AR expressed in AAV-NpyP^+^ neurons may be targets for the development of novel analgesics.

Spinal NA has been previously shown to directly excite a part of inhibitory interneurons in SDH lamina II [23,24,25,34,35], but the subset of inhibitory interneurons responsible for pain modulation by spinal NA remains unknown. In this study, we showed that AAV-NpyP^+^ neurons, a recently identified inhibitory interneuron subset located selectively in lamina IIo [9], mostly respond to NA, and its effect is excitable in most of these interneurons. Indeed, over 60% of these neurons are depolarized by NA. This proportion is higher than that of GAD67-expressing lamina II inhibitory interneurons (approximately 40%) [35], suggesting that AAV-NpyP^+^ neurons are major targets of spinal NA. Furthermore, our data obtained from pharmacological and genetic interventions using selective antagonists for AR subtypes and AAV-NpyP^+^ neuron-specific α_1B_-AR shRNA expression indicated that NA directly excites AAV-NpyP^+^ interneurons via α_1B_-ARs. However, the responsible α_1_-AR subtype for the effect of NA on inhibitory neurons in rats may be different from that in mice. Our previous study using mouse spinal cord slices showed that NA increased the frequency of inhibitory postsynaptic currents in substantia gelatinosa neurons via α_1A_-ARs [26,36]. The reason for the difference in the α_1_-AR subtypes involved remains unclear, but some possibilities may be considered: the differences in the species (rat vs. mouse), population of inhibitory interneurons (AAV-NpyP^+^ neurons vs. *Vgat-Cre*^+^ neurons), and the measured responses (depolarization of cell body vs. inhibitory postsynaptic currents in SDH neurons received inputs from NA-responding neurons). Nevertheless, both α_1_-AR subtypes are coupled with the Gq protein, and their effect on neuronal activity is consistently excitatory.

In addition to the excitatory response, NA also produces hyperpolarization in a small number of AAV-NpyP^+^ neurons. A similar hyperpolarizing effect of NA has also been reported in a part of GAD67-expressing SDH interneurons in mice [35]. Notably, the proportion of AAV-NpyP^+^ neurons with hyperpolarization increased when α_1B_-ARs were blocked or knocked down. The occlusion of NA-induced hyperpolarization by the α_2_-AR antagonist atipamezole leads to the possibility that some AAV-NpyP^+^ neurons co-express both α_1B_-ARs and α_2_-ARs and that the net activity of AAV-NpyP^+^ neurons by NA stimulation could be determined by the balance between excitatory and inhibitory responses via α_1B_-ARs and α_2_-ARs, respectively. Our data showing that NA caused depolarization in the majority of AAV-NpyP^+^ neurons indicate that the net effect of NA in these neurons is excitatory via α_1B_-AR activation.

Within the SDH, AAV-NpyP^+^ interneurons receive excitatory inputs from Aβ fibers and transmit inhibitory signals to lamina I neurons that project to the brain [9]. The role of AAV-NpyP^+^ neurons in spinal pain transmission and processing is evident in neuropathic pain conditions. After PNI, the RMP of AAV-NpyP^+^ neurons deepen, and excitability is impaired. This alteration causes lamina I neurons to be excited by Aβ fiber stimulation, thereby leading to neuropathic allodynia. Conversely, chemogenetically depolarizing AAV-NpyP^+^ neurons after PNI suppresses Aβ fiber-derived neuropathic allodynia-like behavior [9]. Thus, a depolarizing stimulus to AAV-NpyP^+^ neurons could suppress neuropathic pain. This study demonstrated that duloxetine suppressed Aβ fiber-evoked neuropathic allodynia-like behavior and that α_1B_-ARs of AAV-NpyP^+^ neurons are necessary for this effect. Considering that duloxetine increases spinal NA levels [17,18,19] and that the excitatory effect of NA on AAV-NpyP^+^ neurons is retained after PNI, excitation of AAV-NpyP^+^ neurons by spinal NA via α_1B_-ARs is considered critical in the therapeutic effect of duloxetine on neuropathic allodynia-like behavior.

NA has been reported to have multiple sites of action in SDH to modulate pain transmission and processing [11,37]. NA inhibits the presynaptic release of glutamate from C-fibers and excitability of SDH neurons via α_2_-ARs [20], although the role of α_2_-AR in the attenuating effect of duloxetine on neuropathic hypersensitivity remains controversial [19,22,38] (also see Introduction). Furthermore, NA also activates SDH astrocytes, a type of glial cells, via α_1A_-ARs, and astrocytic α_1A_-AR knockout enhances the attenuating effect of duloxetine on PNI-induced hypersensitivity [39]. The role of these ARs in the effects of duloxetine on neuropathic allodynia elicited by Aβ fibers will be an important subject for future research.

## 5. Conclusions

We identified AAV-NpyP^+^ SDH neurons and α_1B_-ARs as the critical targets of spinal NA, which contribute to the attenuating effect of duloxetine on neuropathic allodynia-like behavior and can be targets for the development of novel analgesics. As duloxetine is used to treat allodynia and other pain symptoms in patients with neuropathic pain [1,16], this study advanced our understanding of its pain-relieving mechanism and strengthens our view that enhancing the activity of AAV-NpyP^+^ SDH interneurons is a potential pharmacological strategy for treating neuropathic pain.

## Figures and Tables

**Figure 1 cells-11-04051-f001:**
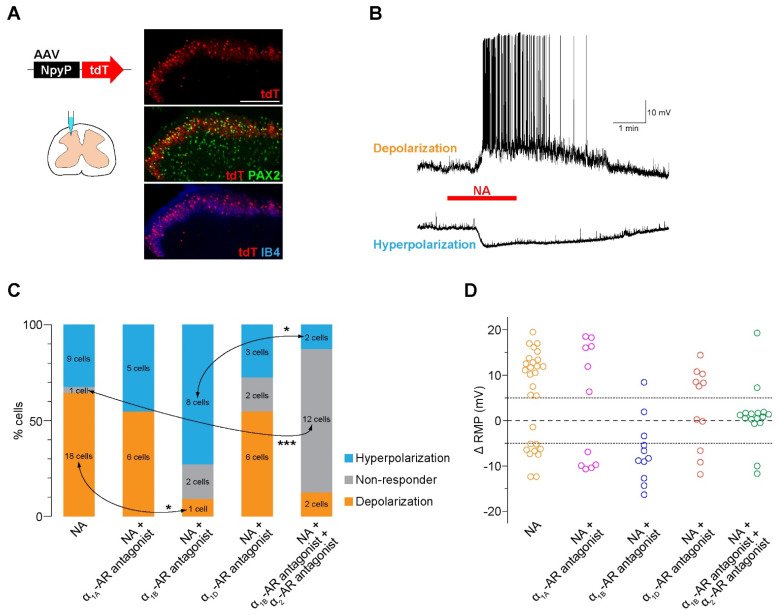
NA excites the majority of AAV−NpyP^+^ neurons via α_1B_−AR. (**A**) tdTomato (tdT) expression on the fourth lumbar (L4) spinal dorsal horn (SDH) 4 weeks after microinjection of AAV−NpyP−tdT. Scale bar, 200 μm. Immunolabeling of tdT^+^ cells (red) with the inhibitory interneuron marker PAX2 (green) or the lamina IIi−selective marker IB4 (blue). (**B**) Representative electrophysiological traces of AAV−NpyP^+^ neurons that were depolarized and hyperpolarized by NA application (20 μM for 2 min). (**C**) Percentage of AAV−NpyP^+^ neurons which responded to NA or NA with each antagonist selective for AR subtypes (*n* = 28 cells from 15 rats for NA, 11 cells from 3 rats for NA + silodosin (α_1A_−AR antagonist), 11 cells from 4 rats for NA + L−765,314 (α_1B_−AR antagonist), 11 cells from 3 rats for NA + A−315456 (α_1D_−AR antagonist), and 16 cells from 6 rats for NA + L−765,314 (α_1B_−AR antagonist) + atipamezole (α_2_−AR antagonist). * *p* < 0.05, *** *p* < 0.001, Fisher’s exact test followed by Bonferroni adjustment. (**D**) Change in resting membrane potentials (ΔRMPs) of AAV−NpyP^+^ neurons before and after NA in slices with or without each AR subtype−selective antagonist treatment.

**Figure 2 cells-11-04051-f002:**
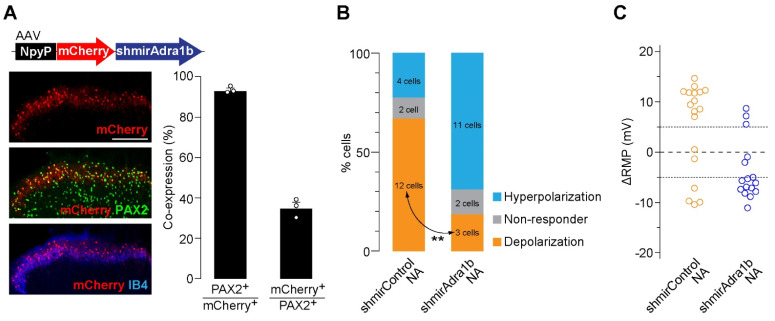
α_1B_-AR in AAV−NpyP^+^ neurons is responsible for NA−evoked depolarization. (**A**) mCherry expression on L4 SDH 4 weeks after microinjection of AAV−NpyP−mCherry−shmirAdra1B. Scale bar, 200 μm. Immunolabeling of mCherry^+^ cells (red) with PAX2 (green) or IB4 (blue). Percentage of co−expressing cells was quantified (680 total mCherry^+^ cells and 1805 total PAX2^+^ cells from 3 rats). Data shown as mean ± SEM. (**B**) Percentage of AAV−NpyP^+^ neurons responded to NA in AAV−NpyP−shmirControl− or AAV−NpyP−shmirAdra1b−microinjected WT rats (*n* = 18 cells from 4 rats for AAV−NpyP−shmirControl, and 16 cells from 5 rats for AAV−NpyP−shmirAdra1b). ** *p* < 0.01, Fisher’s exact test. (**C**) Changes in RMPs (ΔRMPs) of AAV−NpyP^+^ neurons by NA in both groups.

**Figure 3 cells-11-04051-f003:**
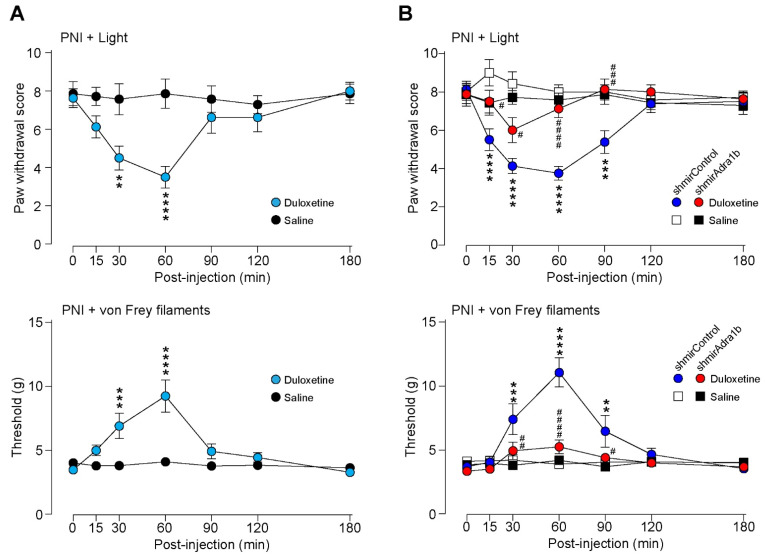
Duloxetine alleviates neuropathic allodynia−like behavior in W−TChR2V4 rats via α_1B_−AR in AAV−NpyP^+^ neurons. (**A**) Paw withdrawal responses to light (paw withdrawal score; see Materials and Methods) and von Frey filaments (threshold) before and after intraperitoneal administration of duloxetine (30 mg/kg) or saline in W−TChR2V4 rats (day 14 post−PNI) (*n* = 7 to 8 rats). ***p* < 0.01, *** *p* < 0.001, and **** *p* < 0.0001 vs. saline−treated group, two−way repeated measures ANOVA with post hoc Bonferroni’s multiple comparison test. Data shown as mean ± SEM. (**B**) Effect of α_1B_−AR−shRNA expressed in AAV−NpyP^+^ neurons on the alleviating effect of duloxetine in neuropathic allodynia−like behavior in W−TChR2V4 rats. Paw withdrawal responses to light (paw withdrawal score) and von Frey filaments (threshold) before and after duloxetine or saline administration in AAV−NpyP−shmirControl− or AAV−NpyP−shmirAdra1b−microinjected W−TChR2V4 rats (on day 14 post−PNI) (*n* = 7 to 8 rats). ** *p* < 0.01, *** *p* < 0.001, and **** *p* < 0.0001, vs. saline−treated AAV−NpyP−shmirControl−microinjected rats, and ^#^
*p* < 0.05, ^##^
*p* < 0.01, ^###^
*p* < 0.001, and ^####^
*p* < 0.0001, vs. duloxetine−treated AAV−NpyP−shmirControl−microinjected rats, two−way repeated measures ANOVA with post hoc Tukey’s multiple comparison test. Data shown as mean ± SEM.

## Data Availability

All data generated or analyzed during this study are included in the paper.

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
