# Peer review of "Identification of Spinal Inhibitory Interneurons Required for Attenuating Effect of Duloxetine on Neuropathic Allodynia-like Signs in Rats"

_cells, 2022, doi:10.3390/cells11244051_

Round 1
Reviewer 1 Report
This study aims to identify spinal inhibitory interneurons required for attenuating effect of duloxetine on neuropathic allodynia–like signs in rats. This is a meaningful, interesting and very well-conducted study that focus on duloxetine on neuropathic allodynia-like signs in rats. Below are a few minor points for the authors’ consideration.
1. This study proved that duloxetine, a spinal cord inhibitory interneuron required for attenuation, has an effect on pain like signs of neuropathic abnormalities in rats, but it is relatively lack of experimental verification. The author needs to supplement the results of action potential conduction of Na+, K+and acetylcholine.
2. Please deeply explore the value brought by the research results, identify the innovation points, and highlight the research results in the discussion section.
Author Response
Re: comment of Reviewer 1
This study aims to identify spinal inhibitory interneurons required for attenuating effect of duloxetine on neuropathic allodynia–like signs in rats. This is a meaningful, interesting and very well-conducted study that focus on duloxetine on neuropathic allodynia-like signs in rats. Below are a few minor points for the authors’ consideration.
Reply: We thank the reviewer 1 for the encouraging comments regarding this study. We have addressed the points raised by the reviewer and modified the text of the revised manuscript.
- This study proved that duloxetine, a spinal cord inhibitory interneuron required for attenuation, has an effect on pain like signs of neuropathic abnormalities in rats, but it is relatively lack of experimental verification. The author needs to supplement the results of action potential conduction of Na+, K+and acetylcholine.
Reply: We think that the reviewer 1’s concern is about the possibility that noradrenaline (NA) activates or inhibits non-AAV-NpyP+ neurons, thereby indirectly altering the electrophysiological state of AAV-NpyP+ neurons. In this study, while we added two blockers for NMDA receptors (D-(−)-2-amino-5-phosphonovaleric acid) and GABAA receptors (bicuculline) in artificial cerebrospinal fluid to block the inputs from other neurons, other blockers such as for acetylcholine receptors were not included. However, we found that a selective α1B-AR knockdown in AAV-NpyP+ neurons abolished the NA-induced depolarization. Therefore, we conclude that NA acts directly on AAV-NpyP+ neurons and produces their excitation.
- Please deeply explore the value brought by the research results, identify the innovation points, and highlight the research results in the discussion section.
Reply: As suggested, we have described the value, key points, and highlights of our study in the discussion section.
Reviewer 2 Report
This study is focused on identifying neurons in the spinal dorsal horn that mediate pain relief from the antidepressant Duloxetine. The authors use an AAV virus to label NPY+ spinal interneurons then investigate the properties of these neurons with respect to drug and or nerve injury using spinal recordings. Overall this is a simple and straight forward, but very high quality, study, that establishes that adrenergic receptor 1B expressed on NPY-expressing inhibitor neurons is required for the antidepressant Duloxetine to provide neuropathic pain relief. This study is a good step towards understanding how to make new “antidepressant” drugs that can effectively treat neuropathic pain.
Minor comments
Figure 1 C-D, the different drugs used are not well known to the general pain community, it could be easier for the reader to understand the results if the authors refer to them by what they block, not what their name is, in the figures, and then include the drug name in the figure legend.
Figure 2a.The significance of the Pzx2/mCherry vs mCherry/pax2 ratio is not clear, can that authors comment on this in the results?
Overall, the authors could try and make the results more accessible to the general audience. For example, what does “delta RMP” measure, what is the significance of this? I understand this is a measure of hyper or depolarization but this could be more easily provided to the reader in the text. Just assume this paper is being read by a pain expert that does not have intense experience with spinal ephys approaches.
Figure 3a. the “score” used should be clearly explained in the results section, please say what this is, how it was calculated, units etc.
Author Response
Re: comments of Reviewer 2
This study is focused on identifying neurons in the spinal dorsal horn that mediate pain relief from the antidepressant Duloxetine. The authors use an AAV virus to label NPY+ spinal interneurons then investigate the properties of these neurons with respect to drug and or nerve injury using spinal recordings. Overall this is a simple and straight forward, but very high quality, study, that establishes that adrenergic receptor 1B expressed on NPY-expressing inhibitor neurons is required for the antidepressant Duloxetine to provide neuropathic pain relief. This study is a good step towards understanding how to make new “antidepressant” drugs that can effectively treat neuropathic pain.
Reply: We thank the reviewer 2 for the constructive comments and suggestions which have helped to improve our study. As described below, we have revised the text and figures to address the points raised by the reviewer.
Minor comments
Figure 1 C-D, the different drugs used are not well known to the general pain community, it could be easier for the reader to understand the results if the authors refer to them by what they block, not what their name is, in the figures, and then include the drug name in the figure legend.
Reply: As suggested by the reviewer 2, we have modified the text in Figure 1C-D and its legend.
Figure 2a. The significance of the Pzx2/mCherry vs mCherry/pax2 ratio is not clear, can that authors comment on this in the results?
Reply: Because we used this vector (AAV-NpyP-mCherry-shmirAdra1b) for the first time, we confirmed that the expression pattern is highly selective to AAV-NpyP+ neurons, as observed in our previous study using the Npy promoter-incorporated AAV vector (Tashima et al., PNAS 118, e2021220118, 2021).
Overall, the authors could try and make the results more accessible to the general audience. For example, what does “delta RMP” measure, what is the significance of this? I understand this is a measure of hyper or depolarization but this could be more easily provided to the reader in the text. Just assume this paper is being read by a pain expert that does not have intense experience with spinal ephys approaches.
Reply: We mentioned this point in the text of the Result section.
Figure 3a. the “score” used should be clearly explained in the results section, please say what this is, how it was calculated, units etc.
Reply: As suggested, we have changed “Score” to “Paw withdrawal score” to make this point more clearly. In addition, because the method for paw withdrawal score measurement was already described in the Materials and Methods section, we have mentioned this in the legend of Figure 3A-B.